# Parental Education and the Association between Fruit and Vegetable Consumption and Asthma in Adolescents: The Greek Global Asthma Network (GAN) Study

**DOI:** 10.3390/children8040304

**Published:** 2021-04-16

**Authors:** George Antonogeorgos, Kostas N. Priftis, Demosthenes B. Panagiotakos, Philippa Ellwood, Luis García-Marcos, Evangelia Liakou, Alexandra Koutsokera, Pavlos Drakontaeidis, Marina Thanasia, Maria Mandrapylia, Konstantinos Douros

**Affiliations:** 1Allergology and Pulmonology Unit, 3rd Paediatric Department, National and Kapodistrian University of Athens, 12462 Athens, Greece; kpriftis@otenet.gr (K.N.P.); drliakouevangelia@yahoo.gr (E.L.); alexandra.koutsokera@gmail.com (A.K.); pauldrakos@hotmail.com (P.D.); marinarod9422@gmail.com (M.T.); marmand24@outlook.com (M.M.); costasdouros@gmail.com (K.D.); 2Department of Nutrition and Dietetics, School of Health Sciences and Education, Harokopio University, 17676 Athens, Greece; dbpanag@hua.gr; 3Department of Pediatrics: Child and Youth Health, Faculty of Medical and Health Sciences, University of Auckland, 1023 Auckland, New Zealand; p.ellwood@auckland.ac.nz; 4Pediatric Allergy and Pulmonology Units, ‘Virgen de la Arrixaca’ University Children’s Hospital, University of Murcia, Network of Asthma and Adverse and Allergic Reactions (ARADyAL) and Biomedical Research Institute of Murcia, IMIB-Arrixaca, 30394 Murcia, Spain; lgmarcos@um.es

**Keywords:** asthma, nutrition, fruits, vegetables, parental, education, adolescents

## Abstract

Background: Evidence suggests that nutritional factors, such as consumption of fruits and vegetables, along with socioeconomic factors such as parental education level, are associated with asthma prevalence. Our study examined the role of parental education in the association between fruit and vegetable consumption and adolescent asthma. Methods: 1934 adolescents (mean age: 12.7 years, standard deviation: 0.6 years, boys: 47.5%) and their parents were voluntarily enrolled and completed a validated questionnaire assessing current asthma status, fruit and vegetable consumption and parental educational level. Participants were categorized as high or low intake for five food groups: fruits, cooked vegetables, raw vegetables, all vegetables (cooked and raw), and all three food groups together (fruits and all vegetables). Results: Adolescents who were high consumers of all three food groups (fruits, cooked and raw vegetables) were less likely to have asthma, adjusted for several confounders (adjusted odds ratio (aOR): 0.53, 95% confidence interval (CI): 0.25–0.97). Moreover, in adolescents who had parents with tertiary education and were in the high consumption of all three food groups, the inverse association was almost twofold higher than the one for adolescents with parents of primary/secondary education (aOR: 0.35, 95% CI: (0.21–0.89) and aOR: 0.61, 95% CI: (0.47–0.93) respectively). Conclusions: Our findings highlight the importance of the adoption of a diet rich in fruits and vegetables for all asthmatic adolescents and emphasize the important role of parental influences in this association.

## 1. Introduction

Asthma is one of the most common non-communicable chronic diseases, with a major public health impact for both children and adults. It is estimated that almost 339 million people are affected by asthma worldwide, while this number could increase up to 100 million more by 2025 [1,2]. Almost one out of five children aged 13–14 years suffers from asthma in English-speaking countries of North America, Europe, and Australasia while an estimated 8.6% of the population between 18–45 years old have experienced asthma-related symptoms (wheezing episodes or whistling breath) in the past 12 months [3]. The burden of the disease has substantial consequences not only for the children but also for the families and the community. Akinbami et al. reported that almost 10.5 million school days and 14.2 million working days were missed during one year in the United States of America due to asthma [4]. In England, 69% of the parents/guardians of asthmatic children reported that they had to take time off their work due to their child’s disease, while 13% lost their job because of the many absences from work [5].

The pathophysiological mechanisms associated with the development of asthma and asthma exacerbations are complex and largely unknown. It is believed that one of the most important underlying factors is oxidative stress. As asthma is a chronic inflammatory disease of the airways, oxidative stress may be implicated in the induction of diverse pro-inflammatory mediators, the enhancement of bronchial hyperresponsiveness, the stimulation of bronchospasm, and the increased secretion of mucin [6,7,8]. Fresh fruits and vegetables are rich sources of antioxidants and elements such as flavonoids, isoflavonoids, and polyphenolic compounds, which can interfere with and minimize the hazardous effect of oxidative stress on asthma. In a recent meta-analysis of 51 observational and 7 experimental studies by Hosseini et al., fruit and vegetable intake appeared to be inversely related to asthma, in both adults and children [9]. In the Greek part of the International Study of Asthma and Allergies in Children (ISAAC)survey, antioxidant food consumption also seemed to be inversely related to asthma development in children irrespective of atopy or heredity [10].

Another important factor that affects childhood asthma is family socioeconomic status (SES), and parental education is one of its determinants. There is a plethora of evidence connecting low SES and other poverty-related exposures with asthma development and management. In a Swedish population-based cohort study, children whose parents were in the lowest income quintile or the lowest educational level had increased risk for childhood asthma and poorer controller medication use than those whose parents had a college degree [11]. Similar findings were reported in an Australian birth-cohort study, which documented the increased risk of asthma at the age of 14 years in children, particularly in girls, who had lived in a low-income family since birth [12]. Moreover, family SES is a major determinant of children’s diet quality. Adolescents with highly educated parents were less likely to adopt unhealthy dietary food patterns such as frequent sweets and fast-food consumption and were more likely to eat fruits and vegetables daily [13,14].

In this study, we used data from the Greek part of the Global Asthma Network (GAN) study, and we aimed to assess four issues: (a) the independent association of high fruit intake, cooked vegetables, and raw vegetables on current asthma (defined as having any wheeze or whistle in the past 12 months; (b) the association of high intake of cooked and raw vegetables combined on current asthma; (c) the association of the high consumption of all three groups with current asthma; and (d) the possible modification of the aforementioned associations by parental education (as a proxy of family SES) in adolescents living in an urban environment in Greece [15].

## 2. Materials and Methods

### 2.1. Design 

This is an observational study, part of the GAN Phase I study, which is an international project aiming to monitor the worldwide prevalence, severity, management, and risk factors of asthma [15].

#### 2.1.1. Setting and Sample

The study took place in the greater metropolitan area of Athens, Greece, from February to March 2020, in high schools; twenty high schools were selected by convenience sampling, from a list that was provided by the Secondary Education Office in Athens. All children in the 1st and 2nd grades of each high school were asked to participate. Schools for children with special educational needs or disabilities were excluded. In total, 2560 child–parent/guardian pairs were asked to participate. Of them, 1934 adolescents, mean age, standard deviation (SD): 12.7 (0.6) years, (921 boys, 47.5% and 1013 girls, 52.5%) and their parents/guardians (25.4% fathers, mean age 49.1 (5.5) years, 74.6% mothers, mean age 45.4 (4.8) years) agreed to participate (participation rate 76%).

#### 2.1.2. Bioethics

The study was approved by the ethics committee of the National and Kapodistrian University of Athens (decision number: 214/13-12-19). For the accomplishment of the study, permission was issued by the Ministry of Educational Affairs (decision number: 10053/24-01-2020).

#### 2.1.3. Measurements

The GAN study included two standardized questionnaires: one that was completed by the adolescents during school time and one intended for parents/guardians to complete at home (adult questionnaire). The adolescents’ questionnaire included several questions about symptoms of asthma as well as other questions regarding their home environment and their lifestyle [15]. Specifically, current asthma was defined as the positive answer to the question “Have you had wheezing or whistling in the chest in the past 12 months?” Moreover, adolescents were asked if they had any siblings, and if so, how many. Adolescents’ parents or guardians were also asked to report if they had a history of atopic diseases (asthma, eczema, or hay fever) if there was visible moisture or mold spots on the walls or ceiling of their homes and whether they smoked. The participating parental/guardian educational level was recorded in three categories (primary (compulsory education/9 years), secondary (non-compulsory/3 years), or tertiary (university/college/postgraduate studies)). Due to the small number of parents who had only primary education (*n* = 24), we merged this level with the next one (secondary education), creating a dichotomous variable (i.e., parental primary/secondary educational level vs. tertiary). The GAN questionnaire included a validated 22-item Food Frequency Questionnaire (FFQ) assessing the past 12-month consumption frequency of 22 food groups or food items [16]. More specifically, adolescents answered 22 questions related to the consumption frequency of 10 food groups; namely, meat, seafood including fish, fruits, cooked vegetables (green and root), raw vegetables (green and raw), pulses (peas, beans, lentils), cereals, dairy (cheese and yogurt), sugar (including lollies/candies/sweets), fast food (excluding burgers), and fizzy or soft drinks, and 12 food items (bread, pasta, rice, margarine, butter, olive oil, milk, eggs, nuts, potatoes, fast food (burgers), choosing one of the three following options: never or only occasionally, once or twice per week, and most of all days for the past 12 months. The consumption of fruits, and cooked and raw vegetables were further recoded into two categories, in order to classify adolescents into high (most or all of the days in the past 12 months) vs. low (never or up to twice per week in the past 12 months) fruits and vegetables intake category. Furthermore, we categorized adolescents into two more consumption groups, according to the combined consumption of the aforementioned food groups. Specifically, in the all vegetables (cooked and raw) consumption group, the adolescents who reported high consumption of both cooked and raw vegetables were classified into the high category, or, otherwise, were labeled as low; and in the fruits and all vegetables consumption group the adolescents who reported high consumption in all the three food groups were classified into the high category, or, otherwise, were labeled as low.

The height and the weight of the participating children were measured and children’s body mass index (BMI) was calculated in order to classify them as normal weight, overweight, and obese, using the International Obesity Task Force (IOTF) classification [17]. These cutoff points are based on health-related adult definitions of overweight (25 kg/m^2^) and obesity (30 kg/m^2^) and adjusted to the age and sex of children. In particular, we measured standing height to the nearest 0.1 cm with a Raven Minimeter (Raven Equipment Limited, Essex, UK) after students had removed their shoes, and body weight to the nearest 0.1 kg on calibrated digital scales (Seca, GmbH & Co. KG, Hanover, Germany).

### 2.2. Statistical Analysis

Continuous variables are presented as mean and standard deviation (SD), and categorical variables are presented as absolute and relative frequencies. Logistic regression analysis was applied in order to estimate the adolescents’ odds and the corresponding 95% confidence intervals (95% CI) of current asthma based on the consumption of fruits, and cooked and raw vegetables in five different models: (a) fruit consumption only, (b) cooked vegetable consumption only, (c) raw vegetable consumption only, (d) all (cooked and raw) vegetable consumption, and (e) all three food groups (fruits, cooked and raw vegetables) combined. Several well-known confounders based on the related literature were included in the models; i.e., sex, BMI, parental atopic history, parental smoking, pet ownership, having an older sibling, cooking with fuels, and indoor exposure to dampness and/or mold. Furthermore, the moderating role of parental educational level in the relation between all the previously reported exposures with the adolescents’ current asthma was examined by the inclusion of the corresponding interaction term in each of the five models. Deviance residuals were calculated in order to evaluate all logistic models’ goodness-of-fit. All reported probability values (*p*-values) were based on two-sided tests and compared to a significant level of 5%. STATA 14 software was used for all the statistical calculations (STATA Corp., College Station, TX, USA).

## 3. Results

Demographic, anthropometric, parental, and home environment characteristics of the study adolescents as well as frequency of fruit and vegetable (cooked and raw) consumption during the past 12 months, according to current asthma status, are presented in Table 1. Current asthma was reported by 6.2% of adolescents in total (50.8% boys and 49.2% girls, *p* = 0.726). Asthmatic participants had a significantly higher mean BMI compared to the non-asthmatic ones (*p* < 0.017). Moreover, 33.3% were exposed currently to dampness and/or mold in their indoor home environment (*p* < 0.011). Regarding fruit and vegetable consumption, significantly fewer asthmatic adolescents consumed fruit almost daily over the past 12 months (*p* = 0.029) while no significant association was observed for each type of vegetables consumption separately or combined (all *p* > 0.05). Finally, almost 50% less asthmatic adolescents were high consumers of all three food groups on a daily or almost daily basis (fruits, cooked vegetables, and raw vegetables) as opposed to their non-asthmatic counterparts (7.5% vs. 13.8% respectively, *p* = 0.03). 

In Table 2 the adolescents’ distribution of the aforementioned characteristics is presented according to the fruits, raw vegetables, cooked vegetables, all vegetables (cooked and raw) and all three food groups consumption category (high vs. low). Adolescents who were in the low fruits, low raw vegetables, low all (cooked and raw) vegetables and low fruits and all vegetables consumption category had a significantly higher mean BMI, while significantly higher current asthma rates were observed for the ones in the low fruits and in the low fruits and all vegetables consumption category (all *p* < 0.05). Regarding parental education level, adolescents who had parents with tertiary education had significantly higher rates of high fruits and high raw vegetables consumptions (all *p* < 0.001). Significantly higher rates of low fruits, low cooked vegetables, low all (cooked and raw) vegetables and low fruits and all vegetables consumption were also found in the adolescents who had at least one parent who had ever smoke in his/her lifetime (all *p* < 0.05).

Figure 1 presents the results of five logistic regression models assessing the association of the consumption of fruits (model 1), cooked vegetables (model 2), raw vegetables (model 3), all vegetables (cooked and raw—model 4), and fruits and all vegetables (model 5) with adolescents’ current asthma status. Adolescents who were high consumers of fruits were 32% less likely to have current asthma after adjusting for adolescents’ sex, BMI, parental atopic history, parental smoking, pet ownership, having an older sibling, cooking with fuels, and indoor exposure to dampness or mold (adjusted OR (aOR): 0.68, 95% CI: 0.45–0.97). A non-significant association was observed for cooked, raw vegetables or all vegetables consumption respectively (all *p* > 0.05). Lastly, when the combined consumption of all three food groups was examined (fruits and cooked and raw vegetables), adolescents who were consuming fruits and all types of vegetables most or all of the days in the past 12 months were 47% less likely to have current asthma compared to the ones who were consuming fruits and all types of vegetables never or up to twice per week in the past 12 months (aOR: 0.53, 95% CI: 0.27–0.95).

The test for a potential moderating role of parental education level on the previous examined five logistic regression models showed a significant interaction only between high consumption of all three food groups (fruits and all vegetables) and current asthma (*p* for interaction <0.001). Thus, we proceeded with a stratified analysis according to the parental education category only for this model. Among adolescents who had parents with primary or secondary education, those with higher consumption of fruits and all vegetables were 40% less likely to have current asthma, adjusted for all the reported confounders (aOR: 0.61, 95% CI: 0.47–0.93), while if they had parents with tertiary education, this inverse association was almost twofold higher (aOR: 0.35, 95% CI: 0.21–0.89).

## 4. Discussion

Our study has demonstrated the inverse association of the high consumption of foods rich in antioxidants such as fruits and vegetables (both cooked and raw) on current asthma in Greek adolescents living in the Athens metropolitan area. Adolescents who had consumed fruits on most or all days of the past year were almost 40% less likely to have current asthma and if they were high consumers of both fruits and all type of vegetables (cooked and raw), the likelihood of having current asthma was almost half compared to the adolescents who were low consumers. Moreover, adolescents who had parents with tertiary education and were in the higher consumption category for both fruits and all type of vegetables (cooked and raw) had a twofold less likelihood of having current asthma than those having parents with primary or secondary education. Thus, our findings highlight the importance of the adoption of a diet rich in fruits and vegetables for all asthmatic children and document that this inverse association was further enhanced in adolescents who lived in families with high parental education.

The role of oxidative stress and inflammation of the airways in the pathogenesis of asthma has been intensively studied for a long time. Specifically, oxidative stress could be attributed to the imbalance between either the increased production of reactive oxygen species (ROS) or the diminished antioxidant defensive capacity of the airway cells of the lungs [18]. Asthmatic children have lower serum levels of antioxidants (vitamins C, E, and uric acid) as well as elevated levels of oxidative stress markers like malondialdehyde during exacerbations compared to non-asthmatic subjects [19]. Moreover, glutathione levels—an airway antioxidant that protects the airway from lipid peroxidation—were much lower in the bronchoalveolar lavage of asthmatic patients compared to non-asthmatic controls [20]. It has also been observed that in asthmatic patients, the equilibrium between oxidants/antioxidants could be further deteriorated by low dietary intakes of the antioxidant vitamins C and E, as well as the lower intake of micronutrients such as selenium and flavonoid [21].

Our findings are in line with the results of many previous studies that report a negative association of fruits or vegetable intake with allergy and asthma in children [22,23]. However, there is a lack of consistency in the reported findings. Some studies suggest that only one of the two food groups has a protective effect on asthma symptoms, while others support the beneficial role of the high intake of both food groups. In the study by Lahod et al., only the higher consumption of vegetables was associated with a lower risk of developing asthma, while fruit consumption was associated with decreased eczema symptoms [24]. Similar findings were reported in the four-year cohort study by Kununoki et al. in Japanese school-aged children, who observed a lower prevalence of asthma in the group with increased fruit intake, but no significant effect was established for the vegetable consumption [25]. However, in the study of Chatzi et al. of 690 children aged 7–18 years living in rural Crete, Greece, high consumption of both fruits and vegetables was inversely correlated with wheezing [26]. The same negative association was reported for the consumption of fruits and raw vegetables in the cross-sectional study of 11,000 children living in three Chinese cities [27]. The aforementioned evidence is supported in a few more studies [28,29]. However, to the best of our knowledge, our study is the first to document not only the distinct negative association of one of the food groups (fruits), on adolescents’ asthma but also the significantly enhanced result of the combined high consumption of both food groups (fruits and all types of vegetable).

Thus, the observed higher inverse relation of the high consumption of both fruits and all types of vegetables to current asthma in adolescence rather than the fruit consumption only could be attributed to the higher and abundant intake of antioxidants that takes place when an adolescent consumes both food groups. It can also be attributed to the association between higher fruit and vegetable intake with the greater adoption of a healthier dietary pattern, such as the Mediterranean diet. Thus, it is possible that children who adhere to a dietary pattern with high consumption of fruits and vegetables also follow a dietary pattern characterized by the high consumption of wholegrain cereals, legumes and nuts, moderate consumption of dairy products and fish, and low consumption of meat. Apart from antioxidants, the Mediterranean diet is rich in several polyunsaturated fatty acids (PUFA) and saturated fatty acids (SFA) and there is evidence that these nutrients have a protective role against asthma, both in adults and in childhood [30,31,32]. However, in the study by Barros et al., although higher dietary intakes of n-3 PUFA and SFA were related to increased likelihood of better asthma control, no significant association was observed with the intake of antioxidant vitamins and minerals [33]. Our evidence supports that a food pattern rich in fruit and vegetable consumption, either independently or as a part of a healthy dietary pattern such as the Mediterranean diet, could be inversely associated with asthma in adolescence possibly due to the increased intake of antioxidants. Complex synergistic associations among the various dietary micronutrients in the concept of dietary patterns such as the Mediterranean diet might exist and should be further researched in order to better understand their beneficial role in asthma prevention.

Another important result of our study is the modification role of parental educational level on the association between fruit and vegetable consumption and asthma. It is well known that low SES is related to unfavorable outcomes in many diseases, including childhood asthma [34,35,36]. Moreover, there is a plethora of evidence that supports the harmful influence of low SES on the diet of teenagers. In the systematic review by Hanson and Chen, in which the association between SES and various outcomes such as cigarette smoking, alcohol consumption, marijuana, and diet was assessed, 90% of the related studies revealed that low SES adolescents had an unhealthier diet compared to high SES teens [37]. In a German cross-sectional study of 1272 adolescents aged 12 to 17 years, higher adherence to a “traditional and western” type diet was associated with decreased intake of important antioxidant components such as vitamins A, C, E, K, and folate. Likewise, as reported in the study by Araujo et al., 13-year-old Portuguese adolescents with low educated parents were more likely to follow an unhealthier and poor in antioxidants dietary pattern that was characterized by high fast food and sweets consumption [14]. Therefore, low family SES is linked to poor diet quality in children and adolescents, with low consumption of fruits and vegetables and resultant low intake of very important antioxidant components, which are of great importance in the physiological development of the lungs. As demonstrated in the cohort study by Sdona et al., children with asthma at eight years of age who had a diet with higher total antioxidant capacity, had improved lung function and decreased odds of having low lung function at 16 years [38]. Parental knowledge of the protective role of fruit and vegetable consumption on asthma is of great importance and should be always included in the management of asthmatic adolescents.

### Limitations

The design of the study is cross-sectional, and thus it suffers the limitations of this type of epidemiological study, such as recall and report bias of the requested information on the assessment of current asthma and fruit and vegetable intake. However, the use of a global research methodology and a validated instrument allows comparisons between countries and expands the generalization of the findings. Moreover, only one parent/guardian reported his/her educational level. However, almost 75% of the parents/guardians that participated in the study were mothers and 70% of them had tertiary education. Since women in Greece until recently were more likely than men to have lower education, the large representation of higher educated mothers in our sample reassures that no underreporting of the parental educational level was made. The cutoffs used of the classification of BMI could influence the strength of the obesity–asthma association. However, the use of BMI as a confounder avoids the introduction of such bias. High fruit and vegetable consumption could be part of a healthy dietary pattern such as the Mediterranean diet and interaction with other micronutrients might exist. However, we discussed these discrepancies in the recent bibliography and we have acknowledged this limitation. Finally, every effort was made not to overinterpret the study results.

## 5. Conclusions

Our study not only documented the inverse association between high consumption of fruits with current asthma in adolescents but also revealed the lower inverse association of the combined intake of all three food groups (fruits and all vegetables) on it compared to the consumption of fruits alone. Furthermore, it has been demonstrated that this inverse association was further enhanced for adolescents who were consuming fruits and all types of vegetables most or all of the days in the past 12 months and had parents with a higher educational level than the ones who had parents of primary or secondary education. Health-care providers should always keep in mind that nutritional interventions have an important place in asthma management and prevention and should advise parents and children to adopt a diet rich in fruits and all kinds of vegetables. Furthermore, the role of parents in the management of asthma is of great importance, and health-care policymakers, as well as medical-care providers, should always target parents in order to get the optimum results.

## Figures and Tables

**Figure 1 children-08-00304-f001:**
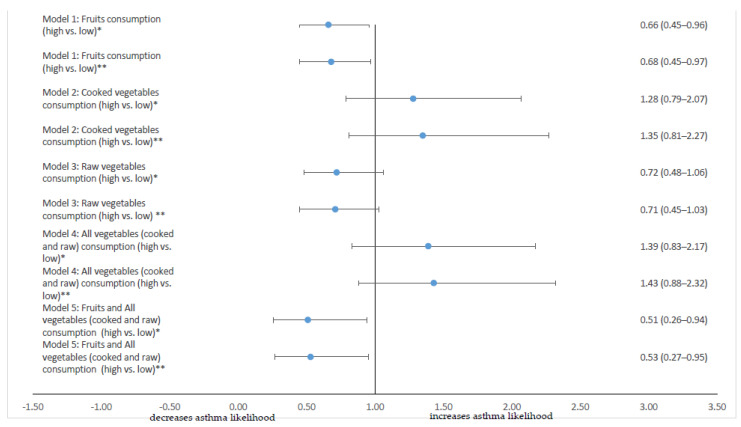
Results from logistic regression analysis (odds ratio (OR), 95% confidence intervals (CI)) assessing the crude and adjusted association of the consumption of fruits (Model 1), cooked vegetables (Model 2), raw vegetables (Model 3), all (cooked and raw) vegetables (Model 4), and fruits and all vegetables (Model 5) on the likelihood of current asthma (defined as having wheezing or whistling in the chest in the past 12 months) on adolescents (*n* = 1934). * Crude OR ** adjusted for: adolescents’ sex, body mass index, parental atopic history, parental smoking, pet ownership, having an older sibling, cooking with fuels, indoor exposure to dampness of mold.

**Table 1 children-08-00304-t001:** Demographic, anthropometric, parental and home environment characteristics and fruit, and cooked and raw vegetable consumption frequency of the participating adolescents according to adolescents’ current asthma status (*n* = 1934).

	Current Asthma *		
	Yes (*n* = 120)	No (*n* = 1814)	*p*
Children’s age (years), mean (SD **)	12.7 (0.59)	12.7 (0.65)	0.51
Children’s BMI (kg/m^2^), mean (SD)	21.7 (4.4)	20.9 (3.5)	0.017
Pet ownership (Yes, *n*, %)	39 (32.5)	524 (28.9)	0.407
Having an older sibling (Yes, *n*, %)	56 (46.7)	773 (42.6)	0.388
Parental atopic history (Yes, *n*, %)	63 (52.5)	795 (43.7)	0.061
Parental ever smoking (Yes, *n*, %)	78 (65.0)	1004 (55.3)	0.038
Parental education level (Tertiary, *n*, %)	75 (62.5)	1207 (66.5)	0.374
Cooking with fuels (Yes, *n*, %)	58 (48.3)	935 (51.6)	0.484
Current exposure to dampness and/or mold (Yes, *n*, %)	40 (33.3)	420 (23.1)	0.011
Fruit consumption frequency (*n*, %)			
*Most or all days in the past 12 months*	69 (57.5)	1219 (67.2)	0.029
Cooked vegetables consumption frequency (*n*, %)			
*Most or all days in the past 12 months*	22 (18.3)	274 (15.1)	0.321
Raw vegetables consumption frequency (*n*, %)			
*Most or all days in the past 12 months*	39 (32.5)	732 (40.4)	0.089
All vegetables (cooked and raw) consumption frequency (*n*, %)			
*Most or all days in the past 12 months*	22 (18.3)	261 (14.4)	0.232
Fruits and all vegetables (cooked and raw) consumption frequency (*n*, %)			
*Most or all days in the past 12 months*	9 (7.5)	251 (13.8)	0.030

* Current asthma: adolescents reporting having wheezing or whistling in the chest in the past 12 months ** SD: Standard Deviation.

**Table 2 children-08-00304-t002:** Adolescents’ demographic, anthropometric, parental and home environment characteristics according to the consumption frequency of fruits, cooked vegetables, raw vegetables, all (cooked and raw) vegetables, and fruits and all vegetables in the past 12 months (*n* = 1934).

	Fruits Frequency Consumption		*p*-Value	Cooked Vegetables Consumption		*p*-Value	Raw Vegetables Consumption		*p*-Value	All Vegetables Frequency Consumption		*p*-Value	Fruits and All Vegetables Frequency Consumption		*p*-Value
	High *	Low **		High	Low		High	Low		High	Low		High	Low	
Children’s age (years), mean (SD)	12.7 (0.6)	12.8 (0.6)	0.02	12.7 (0.6)	12.7 (0.6)	0.912	12.7 (0.6)	12.7 (0.6)	0.431	12.7 (0.6)	12.8 (0.7)	0.312	12.8 (0.6)	12.8 (0.7)	0.734
Children’s BMI (kg/m^2^), mean (SD)	20.8 (3.4)	21.5 (3.7)	<0.001	20.7 (3.6)	21.0 (3.6)	0.158	20.7 (3.4)	21.2 (3.7)	0.005	20.5 (3.3)	21.2 (3.6)	0.015	20.7 (3.4)	21.6 (4.0)	0.015
Pet ownership (Yes, *n*, %)	361 (28.0)	202 (31.3)	0.133	85 (29.1)	478 (29.2)	0.978	214 (27.8)	349 (62.0)	0.294	57 (25.4)	328 (29.3)	0.249	42 (23.0)	63 (32.5)	0.039
Current asthma *** (Yes, *n*, %)	69 (5.4)	51 (7.9)	0.029	22 (7.4)	97 (5.9)	0.321	39 (8.1)	81 (7.0)	0.089	22 (7.8)	98 (5.9)	0.232	9 (3.5)	104 (6.5)	0.030
Having an older sibling (Yes, *n*, %)	546 (65.9)	283 (43.8)	0.562	127 (43.8)	701 (42.7)	0.727	325 (42.2)	504 (43.4)	0.595	101 (45.1)	492 (43.9)	0.733	84 (45.9)	85 (43.6)	0.651
Parental atopic history (Yes, *n*, %)	100 (7.8)	55 (8.5)	0.560	25 (16.1)	130 (7.9)	0.699	68 (8.8)	87 (7.5)	0.291	18 (8.0)	91 (8.1)	0.973	16 (8.7)	18 (9.2)	0.868
Parental ever smoking (Yes, *n*, %)	528 (41.0)	296 (46.0)	0.037	99 (34.0)	725 (44.2)	0.001	317 (41.1)	507 (43.7)	0.266	78 (34.8)	501 (44.7)	0.007	57 (31.1)	93 (47.7)	0.001
Parental educational level (Tertiary, *n*, %)	895 (69.9)	392 (60.1)	<0.001	193 (66.6)	1087 (66.2)	0.918	550 (71.4)	730 (62.8)	<0.001	132 (59.2)	681 (60.6)	0.686	122 (67.0)	116 (59.5)	0.129
Cooking with fuels (Yes, *n*, %)	1195 (93.0)	590 (91.9)	0.385	265 (91.1)	1519 (92.9)	0.268	702 (91.4)	1083 (93.4)	0.094	206 (92.4)	1045 (93.4)	0.584	169 (92.3)	173 (89.2)	0.288
Current exposure to dampness and/or mold (Yes, *n*, %)	299 (23.2)	160 (24.8)	0.438	69 (23.7)	390 (23.8)	0.984	198 (25.7)	261 (22.5)	0.103	55 (24.6)	252 (22.4)	0.491	41 (22.4)	53 (27.2)	0.283

* High: all or most of the days in the past 12 months; ** low: never/occasionally/once or twice a week in the past 12 months; *** current asthma: adolescents reporting having wheezing or whistling in the chest in the past 12 months.

## Data Availability

No applicable.

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
