# Peer review of "Parental Education and the Association between Fruit and Vegetable Consumption and Asthma in Adolescents: The Greek Global Asthma Network (GAN) Study"

_children, 2021, doi:10.3390/children8040304_

Round 1
Reviewer 1 Report
This study is based on the assumption that there is an association between fruit and vegetable consumption and asthma, and further the authors intend to prove, that that association is modified by parental education.
First, I am not entirely convinced about the actual association between fruit and vegetable consumption and asthma. The meta analysis that is referred to in the introduction has many weaknesses, and I don't think the conclusions are very clear. Therefore I think the idea of proving that such association could be due to/confounded by parental education is actually not so bad. The problem is, that this study is flawed in a way that I am not sure the statistics are done correctly.
This is evident from Table 1: It is very confusing to read. First of all, there is missing Ns for the overall groups “Ever had asthma” and “Never had asthma”. It seems that some Ns exceed the total N in those groups, e.g. parental educational level: the N among those with “Ever had asthma” is 1160 which I assume is far more than the total of children with asthma ever (that N is never stated, but I calculate it to be around 172)? In general, the whole Table is built wrong. The percentages in parenthesis should be out of the total “Ever had asthma” and “Never had asthma” and it is the comparison of those percentages, that are the basis of the p-values. I think the whole calculation of Table 1 is wrong, and those are the main results.
Further, the definition of asthma ever is very weak (have you ever had wheeze or whistling sounds in the chest?), and moreover, it is not specified when the asthma was present. At the same time, the fruit/vegetable consumption is based on a questionnaire assessing the consumption in the past 12 months, so there is no certain direction of causality here. The wheezing could have been when the child was 2 years old, and current dietary habits in the 13-year-old students will have nothing to do with that.
Other definitions such as molds in the house is also very weak.
I am not convinced that this study confirmed an association between fruit/vegetable consumption and asthma, and even if the statistics were done correctly, the association is of very little meaning, as there is no timely connection between the dietary habits and symptoms of asthma.
Author Response
This study is based on the assumption that there is an association between fruit and vegetable consumption and asthma, and further the authors intend to prove, that that association is modified by parental education.
Comment 1: First, I am not entirely convinced about the actual association between fruit and vegetable consumption and asthma. The meta-analysis that is referred to in the introduction has many weaknesses, and I don't think the conclusions are very clear. Therefore I think the idea of proving that such association could be due to/confounded by parental education is actually not so bad. The problem is, that this study is flawed in a way that I am not sure the statistics are done correctly. This is evident from Table 1: It is very confusing to read. First of all, there is missing Ns for the overall groups “Ever had asthma” and “Never had asthma”. It seems that some Ns exceed the total N in those groups, e.g., parental educational level: The N among those with “Ever had asthma” is 1160 which I assume is far more than the total of children with asthma ever (that N is never stated, but I calculate it to be around 172)? In general, the whole Table is built wrong. The percentages in parenthesis should be out of the total “Ever had asthma” and “Never had asthma” and it is the comparison of those percentages, that are the basis of the p-values. I think the whole calculation of Table 1 is wrong, and those are the main results.
Reply 1: We would like to thank the reviewer for his/her comments. In compliance with another comment of the reviewer, we reanalyzed out data using as outcome variable the adolescent’s current asthma status, defined as having any as having any wheeze or whilst in the past 12 months, and we totally remade Table 1. Specifically, we removed overall category and in compliance with the next comment we present the distribution of the characteristics of the participated adolescents according to current asthma status with the corresponding number of participants in each group.
Comment 2: Further, the definition of asthma ever is very weak (have you ever had wheeze or whistling sounds in the chest?), and moreover, it is not specified when the asthma was present. At the same time, the fruit/vegetable consumption is based on a questionnaire assessing the consumption in the past 12 months, so there is no certain direction of causality here. The wheezing could have been when the child was 2 years old, and current dietary habits in the 13-year-old students will have nothing to do with that.
Reply 2: We thank the reviewer for his/her comment. Global Asthma Network (GAN) study questionnaire also includes a question related to the reviewer’s comment. Specifically, there is a question assessing the presence of asthma symptoms in the past 12 months. In accordance t to the reviewer’s comment, we used this question as the outcome variable and we procced with new analyses. Thus, we made the appropriate changes in the methods part, and in all the related parts of the Results, Tables and Graphs of the manuscript. More specifically, we changed Table 1, Graph 1, the fourth line in Table 2 and we complete removed Graphs 2a and 2b. Moreover, appropriate changes were made throughout the manuscript, from Abstract to Conclusions.
Comment 3: Other definitions such as molds in the house is also very weak.
Reply 3: We thank the reviewer for his/her comment. Dampness and mold exposure was also self-reported and thus, there is the inherited risk of report bias. However, since there is used as confounder, no important bias could be introduced to our findings.
Comment 4: I am not convinced that this study confirmed an association between fruit/vegetable consumption and asthma, and even if the statistics were done correctly, the association is of very little meaning, as there is no timely connection between the dietary habits and symptoms of asthma.
Reply 4: We thank the reviewer for his/her comment. We hope that the reviewer will be pleased with the revised version of the manuscript, since in the new analyses adolescent’s current asthma status (defined as having any as having any wheeze or whilst in the past 12 months) was used as the outcome variable and no methodological issues regarding the time of exposure are met (dietary habits were assessed the same time period with asthma symptoms).

Reviewer 2 Report
This is an interesting study evaluating the association between fruits and vegetable consumption with reported asthma symptoms in 1934 adolescents from high-schools selected by convenience in an urban area in Greece (greater metropolitan area of Athens). Although the main findings (a protective role of diet rich in fruits and/or vegetables for asthmatic children), are not a novelty the authors also explore the effects of parental education (as a proxy of Socioeconomic status - SES) in these associations, concluding for important role of parental influences in this association. However, it is not quite clear if this results from the SES of the family or an educational influence on nutritional standards.
The main limitation I find in the study is the asthma definition (based on a recall questionnaire) and the several terms the authors derive from the answer to a recall questionnaire within different parts of the text: from “asthma” (title), “asthmatic children” (abstract conclusions, line 32), “asthma in adolescents” (conclusions, line 309), “adolescent’s asthma symptomatology” (Table 1 legend), “asthma ever” (Methods, line 107 & Results, line 164), “ever asthma status” (Results, line 163), “asthmatic adolescents” (Results, lines 169 & 172), “asthma symptoms” (Results, lines 183), “asthma ever symptoms” (Graph1 legend). Considering the lack in the studied cohort of more detailed information such as a medical diagnosis of asthma, lung function tests, medication used (e.g. inhaled beta-2 agonists), the authors should use the same terminology across the manuscript/ tables & graphs legends, and recognize this definition as a limitation of the study.
Considering this observational study is part of a network and international project (GAN, line 84 Material and Methods), and the authors considered the results valid for comparisons between countries and to the generalization of the findings (Discussion, lines 304-306), I was somehow surprised the probable Mediterranean diet pattern and the urban influence on diet patterns of the studied cohort was not touched on the discussion, namely on the possible generalization of findings. Focusing the discussion on the oxidative stress on the pathogenesis of asthma airways inflammation and on the equilibrium of oxidants/antioxidants in dietary intakes (lines 248.-258), the authors should recognize, at least in the discussion, that adherence to a Mediterranean diet pattern, trough other specific nutrients and antioxidants, may influence this relationship and associate with improved asthma control (e.g. Barros R, et al. Dietary intake of α-linolenic acid and low ratio of n-6:n-3 PUFA are associated with decreased exhaled NO and improved asthma control. Br J Nutr. 2011 Aug;106(3):441-50).
Specific points:
Introduction (Lines (L.) 76-81): considering the nature of the study I would suggest to mainly use “assess the association” instead of “assess the effect”:
L. 76 – suggested “a) the independent association of high fruit intake…”
L. 80-81 – suggested “c) the possible modification of the aforementioned associations by parental education (as a proxy of family SES).”
Materials and methods
L. 112. “Their (parents) educational level was orded in three categories (primary, secondary or tertiary).” It is not clear if this relates to the respondent parent/gardian (e.g. 25.4% fathers …74.6% mothers, line 95) or both; this is relevant as the parental education was used as a proxy of Socioeconomic status (aims, line 80). Please discuss and/or consider limitations.
L. 136-138. “…participating children … body mass index (BMI) was calculated in order to classify them as normal weight, over-weight and obese, using the International Obesity Task Force (IOTF) classification”. In the epidemiological literature the strength of the obesity–asthma association in children may be influence by the different BMI classifications adopted (e.g. de Castro Mendes F, et al. Asthma and body mass definitions affect estimates of association: evidence from a community-based cross-sectional survey. ERJ Open Res. 2019 Nov 4;5(4):00076-2019.). Please comment and consider discussing within the limitations/ strengths of the study.
L.149-151. Did the authors use logistic regression? If so, please clarify and how were the confounders selected?
Results:
Table 1. There information regarding the number (N) of participants who had or had not ever asthma is missing. Also, I believe that it is more interesting and improving the reading of the displayed results if the authors present the % according to the column (ever asthma/ no asthma) and not the line.
L. 116-118. “…included a validated 22-item Food 116 Frequency Questionnaire (FFQ)…” Please provide an adequate reference for the validation study, especially as considered by the authors a strength of this study (line 304-306).
L. 181. “…education (all p’s <0.001).” Pleases review as, according to Table 2, only true for Fruits. Vegetables varies with “raw” and “fruits and all vegetables”.
L.192-195. This information should be preferentially stated on the methods section instead of results. Also, please note that the methods section only mentions 3 different models (line 149).
L.199. Please substitute “effect” for “association”.
L. 212-220. If I understood correctly, the parental education was used as an interaction term with the exposure. And if significant than it changed the association between the exposure with the outcome. As such, the information here is not totally clear. Please rewrite: “For instance, among adolescents who had parents with tertiary education, those with higher consumption of fruits were 40% less likely to report ever asthma…(aOR: 0.63…) (Graph 2b),…”. (Please also note that this results is displayed in Graph 2b.)
Graph 2a presents the results from the subgroup with primary/secondary parental education where those with higher consumption of fruits were also 50% less likely to report ever asthma, with an aOR 0.52 (0.22; 0.94). In addition, please clarify how is the interaction significant, if the OR of the association between, for instance, fruits consumption, with ever asthma are on the same direction when you stratified by educational status? [all the participants 0.62 (0.45; 0.87); primary/secondary 0.52 (0.22; 0.94); tertiary 0.63 (0.43; 0.93)]. With these results I would considered that there is no difference between the educational status.
Graph 2a and 2b. if there is only a significant association with two exposures (fruits and fruits + all vegetables), why did the authors stratified for all the exposures? I believe it would be useful “to merge” them into one graph, using only the significant interactions so the readers see clearly the results.
Discussion:
L. 234-247 Opening paragraph: Please state only the main results. In addition, although I agree that the consumption of fruits and vegetables is significantly different between primary/secondary with tertiary (table 2), based on what mentioned previously, the evidence from this study regarding the parental educational level stratification does not clearly support the conclusions that the authors made here. Please review.
Please discuss a possible explanation on why the combination of fruits with vegetables had a greater protective impact on ever asthma.
Limitations:
I believe it would be useful if the authors expand the limitations – for instance, the authors used a self-reported question to define asthma ever (above), the interaction with other Mediterranean diet anti-oxidant properties, generalization, providing an explanation on how they overcame them.
Typos
L. 68-69: “whose parents had s college degree [11]”. I believe “s” is “a”.
L. 95-96 and 145: +/- is missing between the numbers.
L. 112. “orded” consider “was ordered…”
Graph 2a and 2b legends – each as half of the needed information for an easy interpretation without the manuscript; please merge the information and repeat in each Figure.
Author Response
Reviewer: 2
Comment 1: This is an interesting study evaluating the association between fruits and vegetable consumption with reported asthma symptoms in 1934 adolescents from high-schools selected by convenience in an urban area in Greece (greater metropolitan area of Athens). Although the main findings (a protective role of diet rich in fruits and/or vegetables for asthmatic children), are not a novelty the authors also explore the effects of parental education (as a proxy of Socioeconomic status - SES) in these associations, concluding for important role of parental influences in this association. However, it is not quite clear if this results from the SES of the family or an educational influence on nutritional standards.
Reply 1:We thank the reviewer for his/her fruitful comment. Education is a structural part of the SES context and correlates strongly with income1. Moreover, the rationale of the assessment of parental education level in the context of Global Asthma Network (GAN) study is the evaluation of family SES, as indicated by the in the study section of validated instruments2.Since we assessed the parental education level in general and no specific assessment of nutritional parental knowledge was performed, the effect of parental education on the association between fruits and vegetable consumption and adolescent asthma could be attributed to the SES of the family. Therefore, authors feel confident about the use of parental education as a proxy for SES in the context of the Greek part of GAN study.
- Tamborini CR, Kim C, Sakamoto A. Education and Lifetime Earnings in the United States. Demography. 2015;52(4):1383-1407. doi:10.1007/s13524-015-0407-0
- http://www.globalasthmanetwork.org/surveillance/manual/validation.php
Comment 2: The main limitation I find in the study is the asthma definition (based on a recall questionnaire) and the several terms the authors derive from the answer to a recall questionnaire within different parts of the text: from “asthma” (title), “asthmatic children” (abstract conclusions, line 32), “asthma in adolescents” (conclusions, line 309), “adolescent’s asthma symptomatology” (Table 1 legend), “asthma ever” (Methods, line 107 & Results, line 164), “ever asthma status” (Results, line 163), “asthmatic adolescents” (Results, lines 169 & 172), “asthma symptoms” (Results, lines 183), “asthma ever symptoms” (Graph1 legend). Considering the lack in the studied cohort of more detailed information such as a medical diagnosis of asthma, lung function tests, medication used (e.g. inhaled beta-2 agonists), the authors should use the same terminology across the manuscript/ tables & graphs legends, and recognize this definition as a limitation of the study.
Reply 2: We thank the reviewer for his/her valuable comment. Your comment is in compliance with a comment received from another reviewer. Global Asthma Network (GAN) study questionnaire also includes a question related to the reviewer’s comment. Specifically, there is a question assessing the presence of any asthma symptoms in the past 12 months. In accordance to the reviewer’s comment, we used this question as the outcome variable and we procced with new analyses. Thus, we made the appropriate changes in the methods part, and in all the related parts of the Results, Tables and Graphs of the manuscript. More specifically, we changed Table 1, Graph 1 and Graphs 2a and 2b totally, the fourth line in Table 2 as well as all the related part in the Results and Discussion part of the manuscript. Moreover, all the appropriate changes were made in order to keep the same terminology throughout the manuscript and a limitation about the recall bias in current asthma symptoms was made. Pls see lines 362-363.
Comment 3: Considering this observational study is part of a network and international project (GAN, line 84 Material and Methods), and the authors considered the results valid for comparisons between countries and to the generalization of the findings (Discussion, lines 304-306), I was somehow surprised the probable Mediterranean diet pattern and the urban influence on diet patterns of the studied cohort was not touched on the discussion, namely on the possible generalization of findings. Focusing the discussion on the oxidative stress on the pathogenesis of asthma airways inflammation and on the equilibrium of oxidants/antioxidants in dietary intakes (lines 248.-258), the authors should recognize, at least in the discussion, that adherence to a Mediterranean diet pattern, trough other specific nutrients and antioxidants, may influence this relationship and associate with improved asthma control (e.g. Barros R, et al. Dietary intake of α-linolenic acid and low ratio of n-6:n-3 PUFA are associated with decreased exhaled NO and improved asthma control. Br J Nutr. 2011 Aug;106(3):441-50).
Rely 3: We thank the reviewer for his/her valuable comment. According to the reviewer’s suggestion, we added a paragraph in the Discussion and we discuss the correlation of high fruits and vegetable consumption with the Mediterranean dietary pattern and how could this influence the relationship of the aforementioned food groups with asthma. Pls see lines 317-337.
Specific points:
Comment 4: Introduction (Lines (L.) 76-81): considering the nature of the study I would suggest to mainly use “assess the association” instead of “assess the effect”:
- 76 – suggested “a) the independent association of high fruit intake…”
- 80-81 – suggested “c) the possible modification of the aforementioned associations by parental education (as a proxy of family SES).”
Reply 4: We thank the reviewer for his/her valuable comments. All the suggested amendments were made in lines 76-81
Materials and methods
Comment 5: L. 112. “Their (parents) educational level was orded in three categories (primary, secondary or tertiary).” It is not clear if this relates to the respondent parent/gardian (e.g. 25.4% fathers …74.6% mothers, line 95) or both; this is relevant as the parental education was used as a proxy of Socioeconomic status (aims, line 80). Please discuss and/or consider limitations.
Reply 5: We thank the reviewer for his/her valuable comment. The parental education level was reported by the participating parent. The appropriate change was made in line 117-118 and in the limitations part of the manuscript lines 365-370.
Comment 6: L. 136-138. “…participating children … body mass index (BMI) was calculated in order to classify them as normal weight, over-weight and obese, using the International Obesity Task Force (IOTF) classification”. In the epidemiological literature the strength of the obesity–asthma association in children may be influence by the different BMI classifications adopted (e.g. de Castro Mendes F, et al. Asthma and body mass definitions affect estimates of association: evidence from a community-based cross-sectional survey. ERJ Open Res. 2019 Nov 4;5(4):00076-2019.). Please comment and consider discussing within the limitations/ strengths of the study.
Reply 6: We thank the reviewer for his/her valuable comment. We agree with the reviewer about the effect of different BMI classifications on the association between BMI and asthma. However, BMI was used as a confounder in the association and not as an exposure variable. Thus, no significant bias was introduced by the use of IOTF classification cutoffs in our study results. We added the suggested limitation on the Limitations part in lines 371-373.
Comment 7: L.149-151. Did the authors use logistic regression? If so, please clarify and how were the confounders selected?
Reply 7: We thank the reviewer for his/her valuable comment. We applied logistic regression analysis and the choice for the included confounders in all models was based on previously known literature. The appropriate amendments were made in line 156-163 in the Statistical analysis section.
Results:
Comment 8: Table 1. There information regarding the number (N) of participants who had or had not ever asthma is missing. Also, I believe that it is more interesting and improving the reading of the displayed results if the authors present the % according to the column (ever asthma/ no asthma) and not the line.
Reply 8:We thank the reviewer for his/her valuable comment. We have removed the previous Table 1 and replaced it with the amended Table 1. Pls see corresponding Table.
Comment 9: L. 116-118. “…included a validated 22-item Food 116 Frequency Questionnaire (FFQ)…” Please provide an adequate reference for the validation study, especially as considered by the authors a strength of this study (line 304-306).
Reply 9: We thank the reviewer for his/her valuable comment. The GAN study uses a questionnaire designed to assess many allergic, asthmatic and several other associated characteristics, such as home environmental and nutritional exposure, globally and all the related measures has been validated. The necessary reference of the validation of the study instruments has been provided in line 123 and reference #16
Comment 10: L. 181. “…education (all p’s <0.001).” Pleases review as, according to Table 2, only true for Fruits. Vegetables varies with “raw” and “fruits and all vegetables”.
Reply 10: We thank the reviewer for his/her valuable comment. Due to the change on the outcome variable, this sentence was updated with the new Results from Table 2. Pls see lines 193-209.
Comment 11: L.192-195. This information should be preferentially stated on the methods section instead of results. Also, please note that the methods section only mentions 3 different models (line 149).
Reply 11: We thank the reviewer for his/her valuable comment. We removed the sentence from the paragraph and we rephrased appropriately the correspondent part of the manuscript (lines 158-160)
Comment 12: L.199. Please substitute “effect” for “association”.
Reply 12: We thank the reviewer for his/her valuable comment. We made the recommended substitution in line 216.
Comment 13: L. 212-220. If I understood correctly, the parental education was used as an interaction term with the exposure. And if significant than it changed the association between the exposure with the outcome. As such, the information here is not totally clear. Please rewrite: “For instance, among adolescents who had parents with tertiary education, those with higher consumption of fruits were 40% less likely to report ever asthma…(aOR: 0.63…) (Graph 2b),…”. (Please also note that this results is displayed in Graph 2b.)
Reply 13: We thank the reviewer for his/her valuable comment. To further explore for a possible moderation effect of parental education on the association between the consumption of the examined food groups, an interaction term was included in all five examined models. Significant interaction was observed only between parental high consumption of all three food groups (fruits and cooked and raw vegetables). Due to the new analyses suggested by another reviewer, we totally rephrased the paragraph and we used your suggested expression in the report of the new results. Pls see 242-254.
Comment 14: Graph 2a presents the results from the subgroup with primary/secondary parental education where those with higher consumption of fruits were also 50% less likely to report ever asthma, with an aOR 0.52 (0.22; 0.94). In addition, please clarify how is the interaction significant, if the OR of the association between, for instance, fruits consumption, with ever asthma are on the same direction when you stratified by educational status? [all the participants 0.62 (0.45; 0.87); primary/secondary 0.52 (0.22; 0.94); tertiary 0.63 (0.43; 0.93)]. With these results I would considered that there is no difference between the educational status.
Reply 14: We thank the reviewer for his/her valuable comment. Due to the new analyses suggested by another reviewer, no significant interaction of the parental education level was observed in the association between current asthma and fruits consumption. However, the interaction term for the association between fruits and all vegetable consumption remained significant. Thus, according to your suggestion, we report only the effect of the fruits and all vegetable consumption association with current asthma in the two parental education level, primary/secondary and tertiary. Pls see lines 242-254
Comment 15: Graph 2a and 2b. if there is only a significant association with two exposures (fruits and fruits + all vegetables), why did the authors stratified for all the exposures? I believe it would be useful “to merge” them into one graph, using only the significant interactions so the readers see clearly the results.
Reply 15: We thank the reviewer for his/her valuable comment. Due to the new analyses, only one significant interaction was observed, thus we omitted the Graphs 2a and 2b and we reported the results of our analyses. Pls see lines 246-254
Discussion:
Comment 16: L. 234-247 Opening paragraph: Please state only the main results. In addition, although I agree that the consumption of fruits and vegetables is significantly different between primary/secondary with tertiary (table 2), based on what mentioned previously, the evidence from this study regarding the parental educational level stratification does not clearly support the conclusions that the authors made here. Please review.
Reply 16: We thank the reviewer for his/her valuable comment. According to the reviewer recommendations, we updated and modified the paragraph stating only the main results of the study and minimizing the interpretation of the study findings. Pls see lines 257-272
Comment 17: Please discuss a possible explanation on why the combination of fruits with vegetables had a greater protective impact on ever asthma.
Reply 17: We thank the reviewer for his/her valuable comment. In accordance with a previous comment made by the reviewer, we discussed that the observed higher protective effect of current asthma status of both high fruits and vegetables consumption could be attributed to the higher and abundant intake of antioxidants that take place when an adolescent consumes both food groups and also to the association between higher fruits and vegetable intake with the greater adoption of a healthier dietary pattern, such as the Mediterranean diet. Please see lines 317-337
Limitations:
Comment 18: I believe it would be useful if the authors expand the limitations – for instance, the authors used a self-reported question to define asthma ever (above), the interaction with other Mediterranean diet anti-oxidant properties, generalization, providing an explanation on how they overcame them.
Reply 18: We thank the reviewer for his/her valuable comment. According to his/her suggestions we expanded the limitations part of the manuscript. Pls see lines 361-376
Comment 19: Typos
- 68-69: “whose parents had s college degree [11]”. I believe “s” is “a”.
- 95-96 and 145: +/- is missing between the numbers.
- 112. “orded” consider “was ordered…”
Graph 2a and 2b legends – each as half of the needed information for an easy interpretation without the manuscript; please merge the information and repeat in each Figure.
Reply 19: We thank the reviewer for his/her valuable comment. All the appropriate changes were made throughout the manuscript.

Reviewer 3 Report
A very interesting topic, a large studied group, correct methodology, my comments are below: 1. Whether children's diseases such as allergies or lung diseases were the exclusion criteria? 2. Why disability was an exclusion criterion from the study? 3. In line 21, the percentage of the girls is missing. 4. WHY A DIVISION INTO 3 CATERGORIES OF FREQUENCY OF CONSUMPTION WAS MADE? The second category is very broad, it would be better to distinguish also average frequency of consumption. 5.The very long retrospective time used in the FFQ (12 m) poses a risk of errors resulting from forgetting 6. In line 308 it would be better to use” inverse association” instead negative association 7. Whether the relationship between the consumption of specific groups of fruit, such as citrus fruit, etc., was analysed.Author Response
A very interesting topic, a large studied group, correct methodology, my comments are below:
We thank the reviewer for his/her positive comment on our work.
- Whether children's diseases such as allergies or lung diseases were the exclusion criteria?
We thank the reviewer for his/her valuable comment. The Global Asthma Network (GAN) study is a field investigation and no exclusion criteria was set for all the participating adolescents.
- Why disability was an exclusion criterion from the study?
We thank the reviewer for his/her valuable comment. Disability schools were excluded from the original sampling frame due to possible barrier to the completion of the study questionnaire.
- In line 21, the percentage of the girls is missing.
We thank the reviewer for his/her valuable comment. We added the requested percentage. Pls see line 21.
- WHY A DIVISION INTO 3 CATERGORIES OF FREQUENCY OF CONSUMPTION WAS MADE? The second category is very broad, it would be better to distinguish also average frequency of consumption.
We thank the reviewer for his/her valuable comment. Since GAN study is design to assess the prevalence and association of asthma and allergies with several exposure globally, the principal investigators of the study developed and validated this food frequency questionnaire. Although we agree with the reviewer about the board definition of the second category consumption, however in order for our results to remain consistent with the global research methodology, thus allowing the generalization of the findings, we propose to retain the original categorization of the frequency of consumption. We added the related reference. Pls see reference #16
- Global Asthma Network. Validation of Instruments. [cited 2021 28/02]; Available from: http://www.globalasthmanetwork.org/surveillance/manual/validation.php.
5.The very long retrospective time used in the FFQ (12 m) poses a risk of errors resulting from forgetting
We thank the reviewer for his/her valuable comment. We have added a special part in the limitation section of our study about the recall bias in food consumption. Pls see lines 362-363.
- In line 308 it would be better to use” inverse association” instead negative association
We thank the reviewer for his/her valuable comment. In the revised version of our manuscript, we have used the propose phrase in several parts of the manuscript. For example see line 268
- Whether the relationship between the consumption of specific groups of fruit, such as citrus fruit, etc., was analysed.
We thank the reviewer for his/her valuable comment. Unfortunately, no specific information was recorded about the consumption of specific fruits such as citrus, thus were not able to analyze such information.

Round 2
Reviewer 1 Report
I can see that the authors have done a great effort in trying to improve this manuscript according to the reviewers' comments, however, as I read through the manuscript I am making corrections in every 4th line on average, and I don't think it is of sufficient quality for publication.
It seems like the authors do not understand the findings of this manuscript, and they have needed a lot of assistance from the reviewers to analyze the data in the correct way. I simply do not trust the results of this manuscript.
Here is an example of my comments to the current version of the manuscript:
Line 22: delete percentage girls – it is only necessary either percentage boys or percentage girls.
Line 23: delete “adolescent” and just write “current asthma”.
Line 28: delete “50%”
Line 32: replace effect with association
Etc
I almost seems like the reviewers have written a new version of the manuscript, and therefore I don't think it should be accepted for publication.
Author Response
Reply to the reviewer’s comments:
Comment 1: I can see that the authors have done a great effort in trying to improve this manuscript according to the reviewers' comments, however, as I read through the manuscript, I am making corrections in every 4th line on average, and I don't think it is of sufficient quality for publication. It seems like the authors do not understand the findings of this manuscript, and they have needed a lot of assistance from the reviewers to analyze the data in the correct way. I simply do not trust the results of this manuscript.
Response to comment 1: We would like to thank the reviewer for his/her fruitful comment. We agree with the reviewer that his/her contribution, as well as the contribution all the other reviewers, helped authors to significantly address any methodological issues and to substantially improve the quality of their work and we are thankful for this opportunity that they gave us.
The participation of our Department as a center of a global collaboration, the Global Asthma Network, with the use of a common standardized research methodology applied in 353 centers in 135 countries enhance our credibility and provides even greater generability of our study results. One of the main aims of GAN Phase 1 study is to provide new insights in asthma prevalence and its association with several risk factors and we feel that readers would merit from our findings, since the documentation of the role of parental education level as a moderator in the relation between fruits and vegetables consumption is novel and add to the current knowledge about the interweaved associations between nutritional and socioeconomical factors with childhood asthma. In addition, due to the cross-sectional nature of our data, any implication based on our findings about causality was omitted in the revised manuscript.
Furthermore, this revised version of the manuscript was carefully proofread by a native English speaker co-author, and to the best of our knowledge every possible mistake was addressed.
Comment 2: Here is an example of my comments to the current version of the manuscript: Line 22: delete percentage girls – it is only necessary either percentage boys or percentage girls. Line 23: delete “adolescent” and just write “current asthma”. Line 28: delete “50%”. Line 32: replace effect with association
Response to comment 2: We have added this percentage according to the comment of another reviewer during a previous review stage. We omitted the percentage of the girls according to his/her suggestion. We have omitted the word “adolescents”, we have also omitted the noted percentage and we have replaced all indicators of causality throughout the paper.

Reviewer 2 Report
The author´s properly addressed the reviewer concerns and improved the manuscript.
Author Response
Reviewer 2:
Comment 1: The author´s properly addressed the reviewer concerns and improved the manuscript.
Response to comment 1: We would like to thank the reviewer for his/her fruitful comments that helped us to improve our manuscript.
